# Influence of Different Light Spectra on Melatonin Synthesis by the Pineal Gland and Influence on the Immune System in Chickens

**DOI:** 10.3390/ani13132095

**Published:** 2023-06-24

**Authors:** Loredana Horodincu, Carmen Solcan

**Affiliations:** Preclinics Department, Faculty of Veterinary Medicine, “Ion Ionescu de la Brad” Iasi University of Life Sciences, Mihail Sadoveanu Alley, 700489 Iasi, Romania; csolcan@uaiasi.ro

**Keywords:** melatonin, circadian rhythms, immune function, antioxidant, bursa of Fabricius, thymus, spleen, GALT

## Abstract

**Simple Summary:**

The pineal gland, in conjunction with its hormone melatonin, possesses the capability to perceive and interpret the light signals from the environment, thereby regulating the physiology, metabolism, and behavior of avian species. This investigation aims to explore the associations between melatonin and immune system functioning. Through an exploration of the immune-pineal axis, our objective is to gain insight into various aspects, including the development of the pineal gland, the influence of light on pineal secretory activity, and the effects of melatonin on lymphoid organs. The findings of this study demonstrate that the utilization of green monochromatic light (560 nm) and blue monochromatic light (480 nm) leads to an increase in melatonin levels in the bloodstream. Furthermore, it ameliorates the inflammatory response, protected lymphoid organs against oxidative stress, and promotes a stronger immune response. In this case, melatonin should be considered a potent antioxidant and immunomodulator.

**Abstract:**

It is well known that the pineal gland in birds influences behavioural and physiological functions, including those of the immune system. The purpose of this research is to examine the endocrine–immune correlations between melatonin and immune system activity. Through a description of the immune–pineal axis, we formulated the objective to determine and describe: the development of the pineal gland; how light influences secretory activity; and how melatonin influences the activity of primary and secondary lymphoid organs. The pineal gland has the ability to turn light information into an endocrine signal suitable for the immune system via the membrane receptors Mel1a, Mel1b, and Mel1c, as well as the nuclear receptors RORα, RORβ, and RORγ. We can state the following findings: green monochromatic light (560 nm) increased serum melatonin levels and promoted a stronger humoral and cellular immune response by proliferating B and T lymphocytes; the combination of green and blue monochromatic light (560–480 nm) ameliorated the inflammatory response and protected lymphoid organs from oxidative stress; and red monochromatic light (660 nm) maintained the inflammatory response and promoted the growth of pathogenic bacteria. Melatonin can be considered a potent antioxidant and immunomodulator and is a critical element in the coordination between external light stimulation and the body’s internal response.

## 1. Introduction

Animal organisms have evolved in alternation between light and dark environments over millions of years. In order to survive and reproduce, they needed two factors: a favourable environment and an optimal time frame. The meeting of these conditions has led to the synchronisation of all rhythmic changes in physiological processes, including immune processes. Thus, the growth and development of physiological and behavioural functions are influenced by information from the environment, consisting of light, temperature, and magnetic fields.

Biological rhythms are determined by an internal biological clock, which includes the hypothalamus, pineal gland, and retina, which, in turn, are synchronised with light cycles via encephalic, pineal, and retinal photoreceptors. The pineal gland, a neuroendocrine organ existing only in vertebrate species, is involved in the control and synchronisation of several behaviours and physiological processes through the circadian synthesis of its main hormone, melatonin [1]. This organ is found in all classes of vertebrates, from fish [2] to amphibians [3], reptiles [4], birds [5], and mammals [6].

The avian pineal gland represents an important step in phylogenetic evolution when considering aspects of structural changes in pineal cells. Interest in the remarkable differences in the morphology and physiology of the avian pineal organ compared to that of mammals began five decades ago. Phylogenetic and ultrastructural peculiarities have been described by the authors of [7,8,9,10,11,12].

Physiologically, the avian pineal organ is characterised by the fact that melatonin synthesis is related to its endogenous circadian oscillator and its direct sensitivity to light [13,14,15,16,17,18,19,20,21,22,23,24,25,26].

The avian circadian system consists of multiple interconnected systems that are synchronised with the light cycles present in the environment [27].

Gwinner [28] put forward the internal resonance model, which explains the interaction between the oscillator present in the pineal gland and the oscillator in the suprachiasmatic nucleus (SCN), particularly the ventral SCN (vSCN). According to this model, the two oscillators mutually amplify each other by receiving external light information. The pineal gland, through its photoreception abilities, and the vSCN, which receives light input from the retinohypothalamic tract (RHT), secrete a shared periodic signal. This model suggests that these oscillators work in parallel and do not mutually exclude each other.

In birds, the function of the mammalian suprachiasmatic nucleus (SCN) has been associated with two distinct structures: the medial SCN (mSCN) and the visual SCN (vSCN) [29,30,31,32]. These structures are interconnected through a neural network and share the same population of astrocytes. The vSCN contains receptors for melatonin, and the intake of exogenous melatonin at night can inhibit the rhythmic metabolism and electrical activity in the vSCN [33,34,35]. In contrast, during the day, both the mSCN and vSCN are active and synchronised through input from the retinohypothalamic tract (RHT) to the vSCN and possibly extraretinal input to the mSCN. The vSCN outputs a rhythmic regulation of the sympathetic system by releasing norepinephrine (NE) to peripheral systems. One example of this regulation is observed in the pineal gland, where NE inhibits melatonin biosynthesis and release [35].

At the molecular level, avian biological clocks possess a well-conserved genetic network known as clock genes, which are analogous to those found in mammals [36,37,38]. These clock genes exhibit specific patterns of expression throughout the day. Positive elements, such as CLOCK (circadian locomotor output cycles kaput) and BMAL1 (brain and muscle aryl hydrocarbon receptor nuclear translocator-like protein 1), are expressed during the day. Negative elements, including PER 1 and 3 (period) and CRY 1 and 2 (cryptochrome), are expressed during the night. Additionally, clock-controlled genes (CCGs) are downstream targets of these clock genes and control various processes rhythmically [36,39,40,41,42,43]. 

Indeed, positive elements in the avian biological clock, such as CLOCK and BMAL1, are transcribed and translated in the cytoplasm. After translation, they form heterodimers. These heterodimers, composed of CLOCK and BMAL1, then translocate back into the nucleus. Once in the nucleus, they act as transcription factors and activate the expression of negative elements, including PER 2 and 3 and CRY 1 and 2, by binding to specific E-box promoter elements. The transcribed mRNA of these negative elements is subsequently translated in the cytoplasm, leading to the formation of PER 2 and 3 and CRY 1 and 2, which are protein oligomers. These oligomers re-enter the nucleus and interfere with the activity of CLOCK/BMAL1, thereby inhibiting their transcriptional activation. This interplay between positive and negative elements is an essential part of the molecular mechanism underlying the circadian rhythm in avian species [5].

In addition to the primary circadian circuit involving clock genes, there is a secondary circuit that involves two genes: Rev-Erbα and RORα. Both of these genes possess E-box promoters, which play a role in amplifying the circadian cycle by regulating the transcription of the Bmal1 gene [5].

Clock-controlled genes (CCGs) in the avian circadian system include genes that encode enzymes involved in the biosynthesis of melatonin [44].

The genes responsible for encoding key enzymes involved in melatonin synthesis in pinealocytes, namely tryptophan hydroxylase (TPH), arylalkylamine N-acetyltransferase (AANAT), and hydroxide-O-methyltransferase (HIOMT), are subject to regulation by both the molecular mechanisms of the biological clock and direct light-induced effects at the transcriptional and translational levels.

The coordinated and rhythmic interplay between the transcription and translation processes of “positive” and “negative” elements establishes a closed feedback loop that generates circadian oscillations in clock function. This feedback loop also extends downstream to regulate the majority of rhythmic metabolic processes (Figure 1).

## 2. Evolution of the Pineal Gland and How Light Influences Secretory Activity

### 2.1. Structure of the Pineal Gland in Chickens

The avian pineal organ is considered a transitional type between a photosensory organ equipped with photoreceptor cells of lower vertebrates and the mammalian endocrine gland influenced by light stimuli perceived by cells with cones, rods, and a subset of retinal ganglion cells [9]. 

The avian pineal organ is located in an inverted triangular region between the cerebellum and the two hemispheres of the telencephalon. The distal end represents an enlarged portion of the pineal organ, the dorsal wall of which is exposed on the surface of the skull and adheres firmly to the dura mater. The proximal portion of the pineal organ, the pineal peduncle, is connected to the dorsal wall of the third ventricle [12,45].

The pineal organ of the adult chick belongs to type 3, according to Quay and Renzoni’s classification [46]. The histological structure follows a change from the follicular [47] or tubular [48] state of the embryological stage and early maturity to the solid lobular state in the adult [28,29]. A follicle, as described by [47], is characterised by a luminal cell layer specialised in photoreception, being composed of columnar cells and a multistratified parafollicular zone of unspecialised cells that give rise to new follicles. 

In the adult hen, the pineal parenchyma has a compact lobulated appearance under the light microscope. Pinealocytes are clustered in radial arrangements, forming cell rosettes with a hollow, small-diameter centre and irregular margins [47]. The stroma of the pineal gland is denser and more fibrous than that of the young pineal gland due to the progressive fragmentation of the parenchyma after the penetration of thin connective sheets delimiting single rosette territories. Under the electron microscope, the pineal parenchyma of the adult hen shows a solid appearance due to the absence of follicular cavities typical of young animals [47].

According to [49], the pineal gland is populated by three main cell types: photoreceptor-like cells, pinealocytes, and supporting cells. The latter mainly consist of ependymal and astrocyte-like cells. Previous studies have also shown that some cells can differentiate into neuron-like cells in the postnatal mouse pineal gland, and these cells possess dual properties of neurons in the central and peripheral nervous system. Furthermore, neuron-like cells can act similar to interneurons to transmit signals to pinealocytes [50].

Pinealocytes in the avian pineal gland are located among the supporting cells near the luminal surface of the follicles. Photoreceptor-like cells are organised radially around the lumen of each follicle, and their outer segments are modified and different from true photoreceptor cells [9]. 

Depending on the presence and ultrastructure of the outer segments, pinealocytes in the avian pineal gland can be divided into three types: receptor pinealocytes, rudimentary-receptor pinealocytes, and secretory pinealocytes [12,51].

Most avian pineal organs contain some pinealocytes that are modified neuroendocrine photoreceptors. This type of pinealocyte has a cilia-like internal structure that can extend into irregular bulbous structures, a non-synaptic basal neuron, and an abundance of dense-core vesicles storing indoleamine, located mainly in the basal process [52,53,54]. Of the large population of cells of the avian pineal parenchyma, the most clearly defined are the receptor pinealocytes, which constitute a group of cells similar to the pineal photoreceptor cells of poikilotherms. This type of pinealocyte appears to be restricted to follicular pineal organs and is distributed along the edge of the lumen [55]. The main characteristic of this type of photoreceptor cell, which differentiates them from other pineal cell types, is the presence of short outer segments with concentric laminations. These structures were first described by González and Valladolid [56] in *Gallus gallus*, using transmission electron microscopy, showing their existence in the apical area of cells corresponding to the outer segments of photoreceptor cells. The existence and nature of these photoreceptor cells in chickens has also been confirmed in other studies [15,47,57,58,59,60].

The immunocytochemical characterisation of chicken and pigeon pineal glands has shown that photoreceptor-like pineal cells contain molecules that are very similar to those expressed in retinal photoreceptors [61], indicating that chicken pinealocytes are modified or rudimentary photoreceptors. 

The only condition that must be met for a photoreceptive organ to perceive and respond to a light source is that the spectral power, wavelength, and light intensity overlap the spectral sensitivity of its photoreceptor [62]. To stimulate lower vertebrate non-retinal photoreceptors, light must penetrate the overlying tissues. The spectral composition of light reaching photoreceptors is therefore strongly determined by the transmission of overlying tissues [63,64]. 

As the avian pineal gland is located beneath the cranial cage, which is a largely translucent bone [1], light information from the environment reaches the avian pineal gland directly through the photosensitive pinealocytes of the gland. The circadian rhythm in birds is highly synchronised by the pineal gland, retina, and hypothalamus. Melatonin secretion from the pineal gland in the dark is a reflection of the circadian rhythm [65,66,67]. The pineal gland has a complementary role in regulating the rhythmic function of the endocrine system [68], which controls circadian patterns of melatonin biosynthesis and photoreceptors [69,70].

Takahashi and co-workers [22] showed that melatonin synthesis and release from cell cultures in chick pineal glands also exhibit a circadian shift and respond to environmental illumination, suggesting that light perception by the avian pineal gland, through the induction of melatonin synthesis, has a physiological role in regulating function and behaviour in birds [71]. The observation that a direct illumination of chick pineal cultures suppressed the nocturnal increase in N-acetyltransferase activity suggests that the gland contains a photoreceptor. Culture at different wavelength intensities of 400 nm (violet light), 500 nm (green light), 600 nm (orange light), and 700 nm (red light) showed that the wavelength of 500 nm (green light) was the most effective for inhibiting N-acetyltransferase activity [71].

It has been firmly established that the membrane potential of vertebrate and frog pineal retinal photoreceptor cells is depolarised in the dark and hyperpolarised by light stimulation [71]. Therefore, it is possible that a direct illumination of chick pineal cells produces a slow and long-lasting hyperpolarisation of the pinealocyte membrane and prevents nocturnal increases in N-acetyltransferase activity. Such speculation is supported by the finding that depolarizing agents restored the suppression of N-acetyltransferase activity by light, and hyperpolarizing agents prevented the nocturnal increase in enzyme activity [71].

A peptide called visinin, with a molecular weight of 24,000 daltons, was recently discovered in the soluble proteins of the chick retina. Through immunohistochemical analyses, it has been observed that the antiserum against visinin specifically and strongly stains the photoreceptor cells, particularly the cones, in the retina of submammalian species [72,73].

Another study showed that visinin in chick pinealocytes increased significantly after continuous light exposure, indicating the photosensitivity of pinealocytes and a possible role of visinin in photoreception [49,59]. Visinin is the specific marker of cone photoreceptor cells in the chicken retina [49,74,75]. Hao et al. [49] used visinin to identify photoreceptor-like cells in the chicken pineal gland. Following immunohistochemistry, the immunoreactivity of visinin was poorly expressed in the embryonic stage to adulthood, except at 7 days of age, when expression was high, which means that the number of photoreceptor-like cells in the chick pineal gland may be the highest in number at 1 week of age and then gradually decreases. Therefore, it is suggested that visinin-immunopositive photoreceptor-like pinealocytes in the chick pineal gland may play a role in neuroendocrine function [49].

The visual system of birds is more developed than that of mammals because it possesses five types of cones that can sense light information [76]. Red, green, and blue light can be seen by birds because the avian retina possesses cone cells that are sensitive to each of these three colours [77,78]. Therefore, illumination can affect many physiological functions of birds, including the immune system [79,80].

### 2.2. Melatonin Secretion

The pineal gland plays an essential role in regulating and imposing rhythms, from daily to seasonal, in all vertebrates. The most important function of the pineal gland is to produce and release melatonin into circulation. Its production in birds is controlled by three main mechanisms: direct light reception, endogenous reception, and noradrenergic transmission [69,81].

In birds, melatonin is released rhythmically by the pineal gland, with high serum levels at night and low levels during the day [82,83]. Its synthesis and release can be influenced by many environmental factors, such as environmental temperature, magnetic field, and, the most important factor, light [84,85]. Exposure to light at night inhibits melatonin synthesis and secretion [85,86]. Long-term exposure to constant light significantly suppressed circulating melatonin levels in chickens [87,88]. 

Melatonin is one of the important hormones that prevent metabolic and physiological disorders in birds. Melatonin regulates the biological activity of the circadian system, acts on respiration, circulation, excretion, reproduction, and the immune system, and also ensures free radical scavenger activity [69].

Melatonin has the ability to convert environmental information into appropriate endocrine signals. For this reason, melatonin mediates the synchronisation of the physiology, metabolism, and behaviour of animals with optimal environmental conditions. The synchronisation of melatonin secretion with changes in light plays an essential role in the regulation of neuroendocrine processes [89] and in the biology of all cells [90].

Melatonin is a neurogenic hormone synthesised in the dark in the epiphysis and in the light in the retina [91,92,93]. The epiphysis produces about 80% of total serum melatonin [94], which is then transported to other organs via blood circulation and cerebrospinal fluid. It is also secreted by bone marrow, lymphocytes, platelets, the gastrointestinal tract, testes, and skin [95]. The biochemical changes that occur in the 24 h cycle in the body, the production of hormones, and their influence on metabolism are regulated by circadian oscillators in the epiphysis [69,96].

Melatonin participates in numerous physiological processes in the body [97]. Additionally, due to its lipophilic characteristic, it can easily pass through biological membranes. Excess melatonin is inactivated by hydroxylation in the liver [89] and is excreted in the urine [98]. There is a multisynaptic neural pathway linking the pineal gland with the external environment via the retina. Melatonin receptors are not only found in various central nervous systems associated with sensory functions [34], but also in numerous peripheral organs such as the lung [99], spleen [100], and gastrointestinal tract [69,101].

The pineal gland, an endocrine gland, plays a crucial role in secreting melatonin through a two-step process. Initially, the synthesis of key enzymes described earlier coordinates a cascade of chemical reactions, leading to the production of the main hormone in the pineal gland.

In avian pinealocytes, the amino acid tryptophan is taken up from the bloodstream. It undergoes hydroxylation by the enzyme tryptophan hydroxylase (TrH; EC 1.14.16.4), as studied by [102], to form 5-hydroxytryptophan. Subsequently, aromatic L-amino acid decarboxylase (AAADC; EC 4.1.1.1.28) decarboxylates 5-hydroxytryptophan, resulting in the production of serotonin (5HT) [5].

The next step involves the conversion of serotonin to N-acetylserotonin (NAS) by the enzyme serotonin-N-acetyltransferase (AANAT; EC 2.3.1.87), as indicated by Bernard et al. [103]. Finally, NAS is transformed into melatonin through the action of hydroxide-O-methyltransferase (HIOMT; EC 2.1.1.4), as demonstrated by Voisin et al. [5,104].

The significance of key enzymes in the regulation of melatonin biosynthesis has been a subject of debate among researchers. While some scientists emphasise the role of serotonin-N-acetyltransferase (AANAT) as the primary enzyme controlling the rhythm of melatonin production, others, such as Simonneaux and Ribelayga [105], argue that hydroxide-O-methyltransferase (HIOMT) is more crucial as it governs the nocturnal amplitude of melatonin in mammals.

In general, there has been relatively less focus on the enzymes tryptophan hydroxylase (TPH) and aromatic L-amino acid decarboxylase (AADC) in the context of melatonin biosynthesis. However, TPH has been proposed as a key enzyme in the biosynthesis of serotonin, a precursor to melatonin [106].

The varying perspectives on the importance of these enzymes highlight the complex regulatory mechanisms involved in melatonin synthesis. Further research and investigations are necessary to fully comprehend the precise roles and contributions of each enzyme in the biosynthesis and regulation of melatonin [44].

Similar to mammals, the avian pineal gland is also innervated by the sympathetic nervous system through the superior cervical ganglia [45,107].

In contrast to mammals, the regulation of the avian pineal gland through adrenergic signalling exhibits some differences. Avian pinealocytes possess alpha-2 adrenergic receptors, which are activated by norepinephrine (NE) released from sympathetic nerve fibres upon exposure to light. This stimulation of alpha-2 adrenergic receptors during the daytime results in the inhibition of melatonin biosynthesis in the avian pineal gland [26].

The inhibitory effect of alpha-2 receptor activation on melatonin biosynthesis in the avian pineal gland is believed to occur through the inhibition of cAMP adenylate cyclase. This inhibition is mediated by a specific guanine nucleotide regulatory protein called Gi [108]. The activation of alpha-2 receptors leads to the activation of Gi, which subsequently inhibits the production of cAMP, ultimately resulting in the downregulation of melatonin synthesis in the pineal gland [109].

The influence of cAMP on arylalkylamine N-acetyltransferase (AANAT) mRNA expression in the avian pineal gland is relatively minor and is primarily controlled by the pineal clock, which regulates the rhythmic changes in AANAT mRNA levels. However, cAMP does have a direct effect on AANAT enzyme activity. It acts downstream of the rhythmic changes in AANAT mRNA expression, affecting the activity of the AANAT enzyme itself [103].. This suggests that while the regulation of AANAT mRNA expression is mainly governed by the pineal clock, cAMP plays a significant role in modulating the enzymatic activity of AANAT [110] (Figure 1).

Melatonin (N-acetyl-5-methoxytryptamine) is a hormone synthesised mainly by the pineal gland in mammals and birds, as well as in the retina of birds, and plays a major role in circadian and seasonal rhythms [111], including affecting reproduction [112] and the immunity of animals. Melatonin influences many physiological processes by binding to specific transmembrane receptors coupled to G protein. To date, there are three different melatonin receptors in vertebrates: Mel1a, described also as MT1; Mel1b or MT2; and Mel1c [113]. MT1 and MT2 receptor subtypes are present in mammals [114], while the Mel1c subtype has only been identified in fish, amphibians, and birds [115]. In addition, a melatonin-binding site called “MT3” was subsequently characterised as the quinone reductase 2 (QR2) enzyme [116,117]. Moreover, the MT3/QR2 binding site has been shown to be widely distributed in mammals [117]. MT3 has also been shown to be present in the embryonic and post-hatching retina of chickens [118]. Melatonin receptors have also been found in the chick brain [99,119,120] and in numerous peripheral tissues, including chick ovary, lung, spleen, and kidney [99,121,122].

Therefore, melatonin also acts via its nuclear receptors in addition to its membrane receptors. The group of melatonin nuclear receptors is formed by three members: RORα, RORβ, and RORγ [123]. These are part of a family of retinoid-related orphan hormone (ROR) nuclear receptors [124,125]. Nevertheless, the expression models of the three members are specific to each tissue. RORα may be found in the thymus, skin, kidney, muscle, and adipose tissue and may control a wide range of physiological and pathological processes, including the immunological response and the development of the nervous system, circadian rhythm, and oxidative stress. Mel-RORα is crucial for the proliferation of T lymphocytes, autoimmune function regulation [126], and anti-inflammatory effects [123,127]. RORβ primarily resides in the central nervous system and plays a crucial role in regulating circadian rhythm and the formation of sensory organs within the nervous system [102]. On the other hand, RORγ exhibits a similar expression pattern to RORα and is predominantly present in immune cells, muscle, adipose tissue, liver, and kidney [128]. Furthermore, RORγ exhibits a similar expression pattern to RORα and is predominantly present in immune cells, muscle, adipose tissue, liver, and kidney [129]. The importance of RORγ lies in its indispensable role in the development and functionality of immune cells. Previous research has demonstrated that RORγ acts as a critical regulatory factor in the differentiation and maturation of Th17 cells [122] and ILC3 cells [123,130].

RORα is widely expressed in a variety of tissues and in the immune system, where its expression has been observed in both lymphoid and murine myeloid tissue, including different subsets of T cells, B cells, and monocytes [131,132]. Melatonin induces a decrease in RORα levels in the nucleus [133], but cAMP mediates melatonin function and increases RORα expression [134]. RORα has been shown to suppress cell apoptosis and oxidative stress by melatonin in myocardial ischemia [135]. Moreover, levels of TNF-a and IL-6 produced by mast cells and macrophages were increased in RORα^−^/^−^mice (absence of RORα gene) [132]. Interleukin-2 (IL-2) plays an important role in regulating lymphocyte proliferation and development [136] (Liao et al., 2013). Garcia-Mauriño demonstrated that melatonin, through nuclear receptor signalling, regulates IL-2 and IL-6 expression in PBMCs [137]. However, the role of the ROR site remains controversial [138]. It is possible that melatonin affects RORα but only indirectly [127,139], and there are even studies reporting that RORα is not a receptor for melatonin [140,141,142]. They considered that sterols and their derivatives may interact with the ROR binding domain and regulate ROR transcriptional activity. Agez et al. also found that melatonin had no effect on RORα mRNA expression in the suprachiasmatic nucleus of the rat hypothalamus [125,143].

The activity of serotonin N-acetyltransferase, the key enzyme in melatonin synthesis [144,145,146], exhibits a circadian rhythm and responds to direct illumination in isolated chick pineal glands [71]. Melatonin synthesis is rhythmic and driven by an endogenous circadian clock, which is also regulated by environmental photon input. Hydroxyindole-O-methyltransferase (HIOMT), the last enzyme in the melatonin biosynthesis pathway, is present in modified photoreceptor-like cells, whereas pinealocyte-like cells are HIOMT positive only after hatching [49,147] (Figure 1).

Using radioimmunoassay (RIA) technology for melatonin [148,149], melatonin content in the pineal gland and its concentration in the blood have been shown to exhibit daily rhythms with nocturnal growth in chickens [99,150], quail [151], and pigeons [152]. It was reported that, in vivo, the melatonin content of the chick pineal gland was 10-fold higher at midnight than during the day [153].

Some research suggests that the chick pinealocyte immediately after hatching is a photoreceptive endocrine cell and biosynthesises melatonin through direct external photoreception. The amount of immunoreactive melatonin in the pineal parenchyma increases until 21 days of age. Melatonin biosynthesis immediately after hatching was more active in the follicular zone than in the parafollicular zone during the day. These results confirmed the suggestion that the pineal parenchyma as a whole becomes gradually secretory with advancing age after hatching [153].

Chicks exposed to long periods of lighting have larger epiphyses than those kept in darkness [144,154,155]. Conversely, prolonged light exposure in ducks is said to cause their epiphyses and decreases in weight, but the follicular cells are taller and have more mitochondria [156,157]. In contrast to chicks kept in light for 14 h a day, those in continuous darkness have a substantially lower volume of parenchymal cells and reduced lipid content (McFarland et al.). In *Passer domesticus* (house sparrow) females subjected to daily increased photoperiods, pinealocyte nuclei were slightly larger [46]. Pineal gland cells of hens reared in long photoperiods (20L:4D) were reported to be larger, contain more small granules, liposomes, and lipid droplets; moreover, the nerve fibres of these birds had fewer core synaptic vesicles than hens which were subjected to short photoperiods (4L:20D) [58]. Lane et al. [158] failed to find any evidence of morphological or cytological differences between the epiphyses of Japanese quail reared for 10 h or 18 h per day in light, and Ralph and Lane [159] observed no structural changes in *Passer domesticus* pineal glands that could be correlated with changes in natural photoperiod [157].

## 3. The Immune Response under the Influence of Melatonin

Melatonin, the most important hormone of the pineal gland, is involved in anti-inflammatory action and is a possible factor in the activation of lymphocytes and antioxidant enzymes [160,161,162]. It has been suggested that melatonin exerts a regulatory effect on animal immune function. Moreover, its biosynthesis and secretion may be modulated by environmental light [163]. 

Lymphocyte development is affected by many factors, such as stress [164], type of medication [165], and environmental information such as light [125,166]. Melatonin modulates a wide range of physiological events that have pleiotropic effects on the immune system through its receptor subtypes, which consist of a membrane receptor (Mel1a, Mel1b, and Mel1c) [167,168] and a nuclear melatonin receptor (RORα, RORβ, and RORγ) [135,140,169]. Among the nuclear melatonin receptors, RORα plays an important role in many physiological processes, especially in those of the immune system [125,170,171].

Melatonin, a neuroendocrine hormone produced by the pineal gland, is involved in regulating both innate immunity [172] and adaptive immunity [138]. Although melatonin primarily functions through membrane receptors, it also possesses the ability to penetrate cell membranes. Melatonin binding sites have been identified in the nucleus of purified lymphocytes from the spleen and thymus [123,173].

In birds, receptors for melatonin in the primary and secondary lymphoid system were reported by Poon et al. [174], and melatonin binding sites in the thymus, spleen, and bursa Fabricius have been shown to have a physiological role in lymphocyte regulation [69,100]. In addition to impacting immune cell activity, melatonin also modulates immune function by influencing the balance of pro-inflammatory and anti-inflammatory cytokines. During the chronic inflammatory phase, melatonin safeguards the body by inhibiting the secretion of inflammation-inducing cytokines such as IL-6, IL-1β, and TNF-α while promoting the release of IL-10 and IL-2, which are protective factors that mitigate inflammation [175]. This ability of melatonin to enhance immune response by modulating cytokine concentrations during inflammation is referred to as the cytoprotective effect [123].

### 3.1. How Melatonin Influences the Activity of Primary Lymphoid Organs

#### 3.1.1. Bursa of Fabricius

The bursa of Fabricius plays a critical role in the normal development of B lymphocytes in birds, providing a microclimate for B lymphocyte maturation, differentiation, and immunocompetence [176]. The relative weight of the bursa could be affected by external factors such as diet or heat stress [177].

Light is an important environmental factor influencing bird behaviour, growth, and health [163]. A number of studies have demonstrated that different photoperiod regimes and light colours can influence cellular and humoral immune responses in birds [178,179,180,181]. Moore and Siopes [179] found that young birds exposed to short and long light periods had a greater cellular and humoral immune response compared to those exposed to a constant photoperiod. In addition to light intensity and light supplementation regime, light colour also plays an important role. For example, different light colours and photoperiod regimes could influence the cellular and humoral immune response in birds [178,179,180,182].

A decade ago, Jin et al. [87] found that green light stimulates pinealocytes and retinal cells to increase arylalkylamine N-acetyltransferase mRNA levels and melatonin secretion levels in broilers. These findings were used to hypothesise that melatonin may play a critical role in modulating the immune function of the bursa in response to monochromatic light in birds [163].

In birds, the target organ of melatonin is the bursa of Fabricius [101], as exogenous melatonin administration has been shown to increase bursal weight, accelerating humoral immune response and cellular communication in turkeys [183].

In chickens, previous studies have shown that melatonin can mediate B lymphocyte proliferation in the bursa of Fabricius via Mel1a and Mel1c receptors [125,184]. Li et al. [163] reported that green light (560 nm) promoted melatonin secretion and also enhanced B lymphocyte proliferation via Mel1c and Mel1a receptors through activation of the cAMP/PKA pathway [185]. Green light (560 nm) illumination enhanced B lymphocyte proliferation in broiler chicks immediately after hatching, as well as LPS and PCNA expression response, better than blue or red light illumination [163]. This finding supports that of Xie et al. [186], who observed that green light (560 nm) can prolong the antibody effective time (prolong the antibody effective time) and cause antibody production immediately post-hatching in broiler chickens. The change in antioxidant capacity may be a pathway by which melatonin mediates monochromatic light and induces B lymphocyte proliferation in the broiler bursa [163].

Physiologically, melatonin doses have been shown to increase gene expression and activity of GSH-Px, GSH-Rd, SOD, and CAT enzymes [168]. A more recent study showed that green light illumination significantly increased GSH-Px, SOD, and T-AOC activities and inhibited iNOS expression, resulting in a reduced MDA content and NO production in broiler bursa compared to other monochromatic light [163] (Figure 2).

To reinforce the results obtained, the same light treatments were also performed on pinealectomised chickens. The results showed that PCNA expression, bursal lymphocyte proliferation, and plasma melatonin levels decreased in all-light treatment groups after pinealectomy [163]. This highlights the importance of the pineal gland as a photoreceptor organ and immune mediator.

The bursa of Fabricius, which is a primary immune organ in the young, undergoes involution with increasing age. Bursa volume peaks at sexual maturity and gradually decreases. The degeneration of the bursa of Fabricius is linked to the apoptosis of B lymphocytes. A previous study revealed that treatment involving green light followed by blue light (560–480 nm) promoted bursa morphological development and induced B lymphocyte proliferation [123]. Interestingly, this research also found a notable decrease in the expression of pro-apoptotic proteins Bax and Caspase-3 within the bursa when the combination of green and blue light (560–480 nm) was used; moreover, the level of anti-apoptosis Bcl-2 protein was higher in the combination of green with blue light, indicating that the association between these two colours could inhibit the apoptosis of bursal B lymphocytes [123].

Previous research also found that green light treatment followed by blue light (560–480 nm) could better induce the proliferation of bursal B lymphocytes [187] and could increase the level of IgG anti-Newcastle disease virus (NDV) and anti-bovine serum albumin (BSA) in plasma [123,182]. This combination of green and blue light (560–480 nm) may promote the morphological development of the bursa and improve humoral immunity in chickens. However, the level of Bax and Caspase-3 proteins in the bursa was the highest in the group subjected to red light (660 nm) [123].

In addition, it was observed that green light treatment followed by blue light (560–480 nm) not only elevated the levels of the anti-inflammatory cytokine IL-10 in the bursa, but also reduced the levels of pro-inflammatory cytokines IFN-γ, TNF-α, and IL-6. Additionally, this light treatment augmented the antioxidant capacity in the plasma. In contrast, red light (660 nm) enhanced plasma proinflammatory levels and increased lipid peroxidation MDA production. The findings suggest that the sequential exposure to green light followed by blue light (560–480 nm) demonstrates a notable improvement in mitigating the inflammatory response within the bursa. This light treatment also aids in shielding the body from oxidative stress, ultimately hindering the apoptosis of B lymphocytes in the bursa. Conversely, the application of red light (660 nm) elevates the secretion levels of IFN-γ, TNF-α, and IL-6, thereby inducing an inflammatory response. Consequently, this imbalance disrupts the equilibrium between B lymphocyte proliferation and apoptosis [123] (Figure 2).

Earlier studies have demonstrated a negative association between the concentration of melatonin in the plasma and the apoptosis of B lymphocytes in chickens. Similarly, there exists a negative correlation between the plasma melatonin concentration and the levels of RORα mRNA and RORγ mRNA. These findings suggest that melatonin exerts an inhibitory effect on B lymphocyte apoptosis by negatively modulating the expression of nuclear RORα and RORγ receptors [123]. The findings of Xie et al. [186] are consistent with these results in that birds exposed to green and blue light (560–480 nm) had the highest ND antibody titres of all groups.

#### 3.1.2. Thymus

Growing evidence suggests that changes in specific immune parameters correlate with plasma melatonin levels [188,189], the release of which is mediated by photoperiod [87,185,190]. Chen et al. [185] found that green light (560 nm) increased circulating melatonin levels and enhanced T lymphocyte proliferation compared to other monochromatic treatments. All this is supported by other previous studies on the benefits of green light on the spleen and bursa of Fabricius in broiler chickens [163,186]. However, Scott and Siropes [180] found different results. They stated that red light (612–617 nm) improved the cellular immune response compared to green light (543–552 nm) and blue light (435–470 nm) after a 15 week exposure period [180]. Most likely, this discordance was due to differences in age, species, and light source. They used 15-week-old turkeys subjected to fluorescent lamp light treatment, whereas Chen used 2-week-old broilers under an LED lamp [185].

Different subtypes of specific receptors are directly involved in light-induced T lymphocyte proliferation, a process mediated in turn by melatonin in the thymus. Two types of G-protein-coupled melatonin receptors, Mel1a (MT1) and Mel1b (MT2), have been identified in the thymus of several animal species [191,192]. However, the Mel1c receptor has also been described and found only in fish, frog, and chicken [185,193]. 

Chen et al. [185] demonstrated that three subtypes of melatonin receptors (Mel1a, Mel1b, and Mel1c) were expressed in the thymus of broilers. Furthermore, melatonin can mediate green light (560 nm)-induced T lymphocyte proliferation via Mel1b and Mel1c receptors in response to ConA, but not via the Mel1a receptor [185].

Pinealectomy abolished the effect of green light (560 nm) on melatonin secretion and reduced circulating melatonin levels and T lymphocyte proliferative activity. However, exogenous melatonin administration enhanced T lymphocyte proliferation [185]. These studies indicated that melatonin may be involved in T lymphocyte development.

In addition to membrane receptors, other melatonin binding sites have been characterised in the thymus, namely nuclear melatonin receptors. RORα plays an important role in many physiological processes, especially in the immune system [170,171]. Immunohistochemical examinations showed that RORα-immunoreactive cells were mainly located in the medullary and cortico-medullary area of the thymus [185]. 

Many studies have reported that RORα is not a receptor for melatonin [140,142], but Xiong et al. [125] found a negative correlation between RORα expression and melatonin levels or T lymphocyte proliferation in the thymus. RORα could negatively regulate green light (560 nm)-induced T lymphocyte proliferation in the chick thymus by regulating IκB phosphorylation, which promotes RELA nuclear translocation as a main effector of NF-kB intracellular pathway [125].

NF-kB, a transcription factor known for its crucial role in regulating the expression of proinflammatory cytokines and other mediators [194,195,196,197,198], consists of homo- or heterodimers formed by five distinct Rel proteins: p65 (RELA), p50 (NF-kB1), p52 (NF-kB2), c-Rel, and RelB. In the absence of stimulation, NF-kB remains inactive in the cytoplasm, bound to inhibitory kB (IkB) proteins. Upon stimulation with various stimuli, IkB undergoes phosphorylation, ubiquitination, and subsequent degradation. This leads to the activation of NF-kB dimers, which then translocate to the nucleus. Within the nucleus, they bind to kB binding sites located in the promoter regions of target genes, thereby initiating the transcription of diverse proinflammatory mediators, such as iNOS and IL-6 [195,198].

A similar report found that melatonin suppressed nitric oxide and IL6 production in murine macrophage-activated LPS cells by inhibiting NF-κB transcriptional activity [161]. IκB was found to block the nuclear entry of RELA, but this effect was reduced if it occurred after phosphorylation [199]. These results indicate that RORα negatively regulates monochromatic light-induced T lymphocyte proliferation by increasing the level of phosphorylated IκB, which releases RELA for nuclear translocation [125]. However, in human aorta, RORα1 negatively interferes with the NF-κB signalling pathway by upregulating IκBα levels [200]. The difference in results may be due to the use of different cell types or treatment methods. 

Xiong et al. [125] demonstrated that chickens reared under red light (660 nm) exhibit a reduced T lymphocyte proliferation activity and an increased expression of apoptosis proteins within the thymus. This suggests that different wavelengths of light have distinct effects on the vitality of lymphocytes, as supported by previous studies [123,181]. A similar finding has also been reported in *Funambulus palmarum* (Indian palm squirrel) [171]. 

Yang et al. [201] reported that RORα levels in LED light were increased compared to those at night in mouse hippocampal neuronal cells; however, RORα mRNA expression was higher at night than during the day in rat pancreas [125,202].

After pinealectomy, consistent with decreased melatonin concentration and T lymphocyte proliferation, an increase in RORα expression was observed in the thymus of chickens under different monochromatic light, but there were no significant differences between the different light treatments of the non-operated and pinealectomy groups [125]. A previous report showed that RORα reversed the effect of melatonin on the antioxidant status in birds [203]. These data suggest that RORα plays a key role in modulating T lymphocyte proliferation under the action of monochromatic light in chicken thymus [125] (Figure 2).

Monochromatic green light (560 nm) illumination may enhance antioxidant capacity in the chick thymus through T lymphocyte proliferation. Some studies have reported an increased immune function of the thymus through an increased organ index and IL-2 concentration [136,204] following the exposure of chickens to monochromatic green light (560 nm). These results confirm previous studies showing that green light enhanced B lymphocyte proliferation in the spleen [125,163] and small intestine [78,205].

### 3.2. How Melatonin Impacts the Activity of Secondary Lymphoid Organs

#### The Spleen 

Photoperiods may have important influences on the immunity of the avian organism [206,207]. As an example, a short photoperiod significantly increased T cell proliferation in the spleen of *Funambulus palmarum* (Indian palm squirrel) [206] and significantly increased splenocyte number and spleen size in hamsters [163,208,209].

In 6-week-old broiler chicks raised under intermittent light, their T and B splenic lymphocytes were more active than those in chicks raised under constant light [178]. Blatchford reported the effects of the intensity of light on the behaviour and immunity of broiler hens and suggested that a higher light intensity could improve health and provide opportunities for increased normal behavioural rhythms [125,210]. In broilers, a greater IgM response was observed at 50 lx than at 5 lx or 200 lx [181,210].

Splenic lymphocyte proliferation was enhanced in vitro when mice were exposed to short days compared to long days [211]. Prendergast demonstrated that the proliferation of circulating splenocytes was also greater in hamster cells exposed to long days compared to those exposed to short days [212]. These studies investigated the effects of light intensity or photoperiod on lymphocytes in the spleen of different species [181].

These studies focused on the effects of photoperiod or light intensity, while fewer wavelength-related studies have been reported. Guo et al. [213], following previous research, found that green light increased melatonin plasma levels, promoted T lymphocyte proliferation in the spleen, and also increased the organ index [181,213].

Similar to these results, green light stimulation around hatching time resulted in a significantly greater weight gain than birds incubated under dark conditions [181,214].

The combination of green and blue monochromatic light could enhance the proliferation of splenic T lymphocytes in broilers compared to single monochromatic light [182]. An increasing number of reports have shown that light colour can induce an immune response in chickens [125].

Previously, Xie et al. [186,215,216] showed that green and blue light enhances splenocyte proliferation and cellular or humoral immune responses better than red light in broilers. These findings suggested that there is a close relationship between optical wavelength and immune status in birds. During the initial growth stage, green light exhibited the highest activity of IL-2 in the spleen. However, as the growth progressed, the activity of IL-2 in the spleen reached its peak under blue light. These observations were reported in previous studies [123,186,216].

At the moment, the mechanism by which monochromatic light acts to enhance lymphocytes is still unclear. Studies have shown, however, that melatonin increases lymphocyte proliferation by activating an intracellular signalling pathway. Yadav et al. reported that melatonin regulates splenocyte proliferation by Mel1b-induced IP3-Ca^2+^ signalling in *Perdicula asiatica* [185].

These findings corroborated previous studies in mice showing that MT2 and not MT1 may be involved in improving splenocyte proliferation, although that study did not detect Mel1c expression [217]. However, this finding is in agreement with the results of Markowska et al., who found that Mel1b is involved in the decreased proliferation of PHA-stimulated chicken splenocytes. The authors explained that melatonin enhanced PHA-stimulated proliferation when mitogen concentration was suboptimal [178,185].

Nevertheless, Drazen and Nelson [217] stated that the Mel1b receptor, but not the Mel1a receptor, may be involved in melatonin-induced splenocyte proliferation in mice, suggesting that lymphocytes that are derived from different sources (e.g., bursal B lymphocytes vs. splenic T lymphocytes) may respond differently to photoperiod or melatonin [185,212].

Using the immunohistochemical technique, the existence of three nuclear melatonin receptors was demonstrated in the spleen. RORα and RORγ are expressed widely in the red pulp and white pulp of the spleen, while RORβ is mainly expressed in the red pulp [181].

The mRNA expression levels of RORα and RORγ in the splenic mass changed according to the light conditions; following red light treatment, they were expressed more intensely than in green light treatment [181]. These results are consistent with another study in which white LED light for 24 h significantly activated autophagy-related genes and increased autophagosome formation in hippocampal neuronal cells (HT-22), all of which were associated with an increased nuclear RORα receptor expression [181,201].

RORγ is expressed in lymphoid tissues and is essential for thymocyte and lymph node development [218]. Mice that are deficient in RORγ have defective thymocyte and lymphoid organ development [181,219]. Several studies have provided evidence that RORγ negatively regulates cytokine secretion by thymocytes [220]. In addition, melatonin treatment reduced RORγ expression in human cancer cells [139]. 

Xiong et al. [181] found that melatonin mediates green light-induced T lymphocyte proliferation in the spleen by downregulating nuclear receptor RORγ expression, but with no effect on B lymphocytes. However, Zhang et al. [123] demonstrated that a monochromatic light combination of green and blue significantly decreased bursal RORγ mRNA levels, and that this nuclear melatonin receptor plays important roles in mediating bursal B lymphocytes. The difference between the two studies on the nuclear RORγ receptor in two functionally and structurally different lymphoid organs (spleen and bursa of Fabricius) may highlight that RORγ is present in young forms of B lymphocytes and then disappears during secondary lymphoid organ population. This assumption is reinforced by Eberl and Littman’s [221] study, which found that RORγ is expressed in the DP stage of T cell development but is absent in mature thymocytes and mature splenocytes. 

## 4. Intestinal Mucosa-Associated Immune System—GALT

The intestinal mucosa is not only the primary site of nutrient digestion and absorption, but also the innate defence barrier against most intestinal pathogens [222,223]. The intestinal barrier is in turn composed of a mechanical barrier, a biological barrier, a chemical barrier, and an immunological barrier [215,224].

### 4.1. Mechanical Barrier

Studies on chickens have suggested that several factors, such as an altered diet or poor husbandry, may influence the intestinal mechanical barrier by altering the depth of the crypts and the height of the villi [225,226]. Light may also be an important environmental factor influencing behaviour, growth, and health in birds.

The height of the villi, the crypt depth, and the ratio between them may also represent the gut mechanical barrier [215,225,226,227]. Morphometric measurements of villus length and intestinal crypt depth are widely used as factors contributing to the preservation of intestinal homeostasis [228]. Shorter villus length and larger crypts may lead to a poorer nutrient absorption and consequently harm animal health [215,229,230].

It needs to be considered that digestion occurs in birds in the upper part of the small intestine, that is, in the duodenum, and released nutrients are mainly absorbed in the lower part of the small intestine. As a consequence, most nutrient absorption occurs in the jejunum and ileum [231], which could be affected when both values, the length of the villi and the depth of the crypts, are reduced.

A previous study by Xie et al. [215] reported that green light and blue light promote better small intestinal mucosal structure, and they revealed that green light resulted in an increase in intestinal villus height and villus-to-crypt ratio and a decrease in crypt depth at an early growth stage (7–21 days) and blue light at a later growth stage (49 days) (Photo no. 3). Similar results have been reported in Arbor Acres broilers [232], Ross 308 broilers [233], and Leghorn chicks [234].

### 4.2. Immunological Barrier

GALT is a key immune system in birds, estimated to comprise more immune cells than any other tissue [235], with associated structures forming a site that promotes the co-localisation of the many types of immune cells required to initiate and mediate immune function. Immune-associated cells in the intestinal mucosa typically contain intraepithelial lymphocytes (iIELs) and IgA^+^ cells, which are the main immune factors [215].

Prior studies have suggested that iIEL is a population of T lymphocytes and may be of particular importance as an immunological barrier during mucosal immune response [215,236,237].

iIELs comprise a heterogeneous population of cells that are located in the basal and apical parts of the epithelium [238,239]. At hatching, a few IELs are detected and their number increases greatly with age; however, in older birds, their number decreases significantly [239].

IgA^+^ cells are the main element of humoral immunity in the gastrointestinal tract [240,241] and are often spread from the crypts to the tips of the villi [215].

It has been shown that sIgA, produced by IgA^+^ cells, is involved in immunological barriers in the gut [242], contributing to the maintenance of intestinal homeostasis [243]. Therefore, the number of sIgA and IgA^+^ cells in the small intestine can be considered indicators for the appreciation of intestinal mucosal immunity [232,234,244].

In an earlier study, Xie et al. [215] reported that green light and blue light improved gut mucosal immunity in broiler chickens, and they revealed that the population of iIEL and IgA^+^ cells became evident in the group subjected to green light treatment at an early growth stage (7–21 days) and blue light treatment at a later growth stage (49 days) [215] (Figure 3).

These results agree with those obtained by Li et al. [78], namely that green light had a positive influence on IgA^+^ cell counts at 14 days of age in the cecal tonsil of chickens. However, the growth induced by light treatment disappeared after pinealectomy, a finding that suggests that melatonin secreted by the pineal gland under the influence of green light is essential for the improvement of immune function [78].

### 4.3. The Chemical Barrier

Melatonin is capable of indirectly enhancing antioxidant activity and removing oxygen free radicals, which in turn regulate the antioxidant status of organs. Physiological doses of melatonin, for example, have been shown to increase gene expression and enzyme activities of GSH-Px, GSHRd, SOD, and CAT [168]. Furthermore, melatonin may also decrease iNOS activity in mice [245]. 

Melatonin is secreted as a function of photoperiod. Various studies have been conducted, focusing on the influence of monochromatic light on antioxidant activity in cecal tonsils. Li et al. [78] demonstrated that monochromatic green light promoted GSH-Px, SOD, and T-AOC activities and inhibited the iNOS-positive cell population and MDA content in 14-day-old chicks. However, the enhanced effect of green light disappeared after pinealectomy, indicating a close relationship between the pineal gland, melatonin, and different monochromatic light [78] (Figure 3).

The chemical barrier of the intestinal mucosa may also be represented by the population of goblet cells, which are single glandular cells distributed along the columnar epithelia [246]. Their role is to secrete high molecular weight glycoproteins known as mucins [247]. Mucus, which is secreted by goblet cells, is an integral structural component of the intestine that acts as a protective environment and participates in maintaining local intestinal immune homeostasis [248], lubrication, and transport between luminal contents and the epithelial lining [249,250,251]. 

So far, 21 different mucin genes have been identified. Of these, Muc2, an essential component for barrier integrity, is highly expressed by intestinal goblet cells [252,253] and plays an important protective role in the gut by preventing foreign bacteria and other pathogens from crossing the intestinal mucosa to come into contact with the cells underlying the epithelial cells [254]. Muc2^−^/^−^mice have bacteria in direct contact with epithelial cells and spontaneously develop colitis and cancer [246,255,256].

Uni et al. [257] reported that the first goblet cells develop in the small intestine of a broiler chick starting 3 days before hatching (at 18 embryonic days) and contain only acidic mucins at that time. In comparison, Yu et al. [246] reported that goblet cells in the small intestine first appear at 15 embryonic days, which is approximately 3 days earlier than the time reported by Uni et al. [257]. This discrepancy in the timing of goblet cell development is most likely attributed to differences in materials and methods used in the studies [246].

Previous studies have shown that the number of intestinal goblet cells is significantly increased when broilers are reared under monochromatic green light at an early age and under monochromatic blue light at an older age [215]. This research is similar to that of Yu et al. [246], in which they proved that monochromatic green light can alter the proportion of goblet cell types and promote the proliferation and maturation of goblet cells in broilers during late embryonic stages. This pattern is similar to that found in other previous studies [246,257] (Figure 3).

### 4.4. The Biological Barrier 

The intestinal epithelium serves as a vital natural barrier, safeguarding against the entry of pathogenic bacteria and toxic substances present in the intestinal lumen. However, various stressors such as pathogens, chemicals, and other factors can disrupt the balance of the normal microflora or impair the integrity of the intestinal epithelium. Consequently, the permeability of this natural barrier can be compromised, thereby facilitating the invasion of pathogens and harmful substances. This disruption in the intestinal barrier function can have diverse effects, including alterations in metabolism, impaired nutrient digestion and absorption, and the initiation of chronic inflammatory processes within the intestinal mucosa [258,259].

Stress is an important factor that makes birds vulnerable to potentially pathogenic microorganisms such as *E. coli*, *Salmonella* spp., *Clostridium* spp., *Campylobacter* spp., etc. This pathogenic microflora in the small intestine competes with the host for nutrients and also reduces the digestion of fats and fat-soluble vitamins due to bile acid deconjugation effects [260].

A change in the gut microbiota of chickens may influence their immunity and health. However, changes in the gut microbiota of chickens can be influenced by several factors. These factors include maintenance conditions, feed composition, and the presence of antibiotics in animal feed [261].

Recent studies have been conducted to better understand the effects of different combinations of monochromatic light on changes in the microbial population in the broiler cecum [262].

Zhang et al. [205] observed that chickens subjected to monochromatic white light treatment had the highest population of *Lactobacillus* spp. In a previous study, it was predicted that *Lactobacillus* spp. could efficiently ferment carbohydrates and inhibit the colonisation of other bacteria by lowering pH levels in the digestive tract [205,263]. Lactic acid bacteria are valued as beneficial bacteria by producing acids (lactic acid) and bacteriocin-like substances [262,264].

Short-chain fatty acids (SCFAs) are the main metabolites produced by the microbiota in the large intestine through the anaerobic fermentation of indigestible polysaccharides, such as dietary fibre and resistant starch [265]. *Butyricicococcus* spp., *Ruminiclostridium* spp., and Ruminococcaceae [266,267,268] are SCFA-producing bacteria, and their populations were enriched in chickens subjected to a monochromatic colour combination of green and blue light, which could be responsible for the high level of SCFA in cecal tonsils in chickens [123,205].

*Butyricicococcus* spp. can produce a significant amount of butyrate, which has important immunomodulatory functions and acts as a modulator of chemotaxis and immune cell adhesion [269]. Butyrate, as a modulator of gut mucosal immunity, can regulate IL6 [270] or TNF [271] secretion, activate B lymphocytes to promote antibody production, and induce T lymphocyte proliferation [263]. In addition, a larger population of the genus *Faecalibacterium* has been characterised as an anti-inflammatory mediator [205,272].

The most negative consequences were found in the case of red monochromatic light treatment, as it promoted the growth of pathogenic bacteria such as *Escherichia*–*Shigella* [205]. Recently, studies have found that *Escherichia*–*Shigella* has been recognised as negatively correlated with fat growth and digestibility [273] and has become a significant cause of morbidity and mortality in broilers [205,274].

The consideration of melatonin secretion by enterochromaffin cells (ECs) in the digestive system is important. Significant progress has been made in understanding the localisation and physiological functions of melatonin in the gastrointestinal tract (GIT) since its discovery [275]. Numerous reviews have explored various aspects of melatonin in the GI tract [276,277,278,279,280].

Bubenik et al. [281] utilised specific melatonin antibodies to localise melatonin throughout the entire GIT of rats. Their study demonstrated that melatonin distribution in the GIT corresponds to the presence of serotonin-producing EC cells [281,282,283], supporting the hypothesis that melatonin is produced from serotonin in the GI mucosa. Subsequent studies employing radioimmunoassay (RIA) and high-performance liquid chromatography (HPLC) confirmed the presence of melatonin in the GIT [284].

Several authors, particularly in birds, reported a significant diurnal variation in melatonin concentrations in certain GI tissues, with higher plasma levels during the night [284,285,286]. Pinealectomy in pigeons reduced midnight melatonin levels by approximately 50% but did not affect midday levels [284]. Conversely, in rats, pinealectomy lowered melatonin levels in serum but had no impact on GI tissue levels [287]. While pinealectomy attenuated nighttime melatonin levels in blood, it had no effect on daytime levels, which are likely produced in the GIT [95,209,288,289,290].

The injection or oral administration of exogenous melatonin and its precursor tryptophan led to a significant accumulation of melatonin in the GIT [285,288,291,292]. The GIT significantly contributes to circulating plasma levels of melatonin, particularly during the daytime [285,293,294,295,296].

The rapid increase in melatonin in the blood following re-feeding [297] may potentially trigger a speculative pulse that influences the shift in biological rhythms. Changes in locomotion, temperature, and cortisol rhythm have been observed after food intake. Notably, these rhythm shifts were not mediated by the pineal gland, as they persisted even after lesions in the suprachiasmatic nucleus, which regulates the circadian rhythm of pineal melatonin synthesis [298].

Conversely, pinealectomy or sham pinealectomy had no effect on the reported changes in sleep patterns, suggesting an independent effect unrelated to the pineal gland [299]. These circadian activity changes related to food intake could be attributed to a pulse of melatonin released from the GIT, which initially enters the peripheral circulation and subsequently affects the circadian clock [295,297].

The considerable variations in melatonin concentrations in GIT tissues appear to be influenced by food intake rather than photoperiodicity, unlike pineal-produced melatonin [293,297,300]. Species differences may reflect diurnal and nocturnal tendencies as well as monophasic or polyphasic feeding behaviours. Additionally, variations in digestion occur between monogastric and polygastric animals [280,301,302].

Immunohistology and RIA have predominantly localised melatonin in the mucosa of the GIT [275,281,286,296,301,303], including the intestinal villi [286,303]. Conversely, lower concentrations and bindings of melatonin have been detected in the submucosa and muscularis [277,286,300,303]. These findings support the speculation that melatonin is produced by enterochromaffin cells in the mucosa but can act as a paracrine hormone in other segments of the GIT [277].

The effects of melatonin synthesised in the gastrointestinal tract primarily occur through paracrine mechanisms, while pineal gland-derived melatonin exerts changes through humoral and neurocrine pathways. In addition to its biorhythmic, antioxidant, and immunomodulatory effects, melatonin influences motor functions, microcirculation, and mucosal cell proliferation in the gastrointestinal tract.

Melatonin secreted in the digestive tract is not regulated by the pineal gland but contributes to overall serum melatonin levels. Together with melatonin from the pineal gland, they synergistically affect the immune system through the aforementioned mechanisms.

## 5. Influence of the Activated Immune System on the Pineal Gland 

The immune and neuroendocrine systems maintain a reciprocal relationship, where they communicate with each other through soluble factors to regulate various functions. Soluble factors, such as cytokines, produced and released by the immune system, can affect the activity of the pineal gland. This interaction completes the information cycle required for the maintenance of homeostasis [304].

The avian embryo presents a valuable model for investigating the communication between the pineal gland and the immune system. This is because the pineal gland and thymus, which are both vital components of these systems, initiate their development simultaneously [304,305,306]. Through the removal of the pineal gland (pinealectomy) in embryos at 96 h of incubation, Jankovic et al. [307] observed a noticeable delay in the development of the thymus and bursa of Fabricius, which are primary lymphoid organs. Additionally, they noted a decline in both the humoral and cellular immune responses. These findings suggest that an intact pineal gland is necessary for the normal progression of the immune system’s development. Furthermore, they indicated that the pineal gland potentially exerts its influence on lymphoid organs either directly or indirectly through other secretions of the neuroendocrine system [306]. 

Another interesting finding related to pinealectomy is that when performed on chicks shortly after hatching, it resulted in the disruption of the circadian rhythm of immune activity. However, this effect was reversed when the chicks were subjected to extended treatment with extremely low doses of melatonin, resembling the natural secretion process [308]. This suggests that melatonin plays a crucial role in regulating the circadian rhythm of immune function and highlights the significance of the pineal gland in maintaining this rhythm.

Further evidence of the connection between the pineal gland and the immune system comes from studies involving the bursa of Fabricius, a primary lymphoid organ specific to birds involved in B lymphocyte maturation. Youbicier-Simo and his team [309] conducted experiments where they removed the bursa of Fabricius during early embryonic stages. They found that this bursectomy not only resulted in a decreased immune response in the chicks but also had an impact on the circadian rhythm of the pineal gland. Specifically, the nocturnal peaks in pineal AANAT enzyme activity and serum melatonin levels were reduced [309]. These findings suggest that the bursa of Fabricius plays a role in regulating the pineal gland’s circadian rhythm and reinforces the interplay between the immune system and the pineal gland.

In order to establish a complete regulatory loop between the immune system and the pineal gland, the activated immune system needs to transmit messages that can be interpreted by the pineal gland, leading to the activation of melatonin synthesis. Several studies have provided evidence supporting this hypothesis. Cytokines, particularly those with endogenous opioid-mediated effects, have emerged as the most promising candidates for fulfilling this function [310]. These findings suggest that cytokines, possibly with the assistance of endogenous opioids, play a crucial role in transmitting signals from the immune system to the pineal gland, ultimately influencing melatonin production.

In addition to their role in immune function, leukocytes have the ability to produce hormones and neuropeptides [311]. One hormone that plays a significant role in neuroimmunomodulation is melatonin, which has shown pronounced effects on the neuroendocrine system. Interestingly, melatonin production has been observed in leukocytes [312,313,314,315], suggesting its involvement in maintaining immune homeostasis [304,316,317]. A study by Finocchiaro et al. [313] demonstrated that peripheral blood mononuclear leukocytes (PBML) can convert serotonin (5-hydroxytryptamine (5-HT)) into melatonin, highlighting the potential existence of a closed circular pathway within the neuroendocrine system. Moreover, there appears to be an immunoregulatory circuit involving indoleamine, as IFN (interferon) stimulates the production of serotonin and melatonin by macrophages and lymphocytes, while these indoleamines inhibit the synthesis of IFN [100,318]. These findings indicate the intricate relationship between the immune system, indoleamines, and melatonin, suggesting a complex interplay within the neuroimmunomodulatory network [319]. 

In a study conducted by Turkowska and colleagues [320], diurnal variations in the mRNA levels of IL-6 and IL-18 were observed in peripheral blood leukocytes (PBL) of chickens. These fluctuations in mRNA levels were partially reversed when the chickens were exposed to constant light conditions. However, when melatonin was added to their drinking water, the diurnal changes in IL-6 and IL-18 mRNA levels were restored [320]. This suggests that melatonin supplementation can regulate the diurnal fluctuations of these cytokines in PBL, highlighting the role of melatonin in modulating immune responses in chickens [44].

Further evidence of feedback from the immune system to the pineal gland is provided by reports of lymphocyte accumulation within the pineal gland in both avian and mammalian species [321,322,323,324,325,326,327]. This accumulation of lymphocytes forms what is known as pineal-associated lymphoid tissue (PALT). In chicks, PALT can constitute up to 30% of the total volume of the pineal gland [321]. The extent of lymphocyte infiltration and the size of PALT suggest the existence of novel mechanisms underlying neuroimmune interactions. PALT is typically found along the outer margin of the pineal gland, but it can also be located centrally between follicles. The population of PALT is characterised by significant accumulations of mononuclear cells that distinguish them from the surrounding pineal tissue. These findings provide additional support for the intricate communication and interplay between the immune system and the pineal gland.

Cogburn and Glick [328] conducted studies demonstrating that the size of pineal-associated lymphoid tissue (PALT) is dependent on the presence of the bursa of Fabricius and the thymus. They observed a reduction in PALT size following bursectomy and/or thymectomy. In a subsequent study [329], they found that tritium-labelled lymphocytes from the thymus and bursa migrated to the pineal gland but only in pups older than 4–5 weeks. These findings have sparked interest in investigating whether the chick pineal gland contains soluble chemotactic factors that can induce the migration of specific subtypes of lymphocytes.

Immunological mechanisms play a role in regulating the neuroendocrine functions of the pineal gland, and this involves the action of immune-derived cytokines that modulate neuroendocrine activities. The trafficking of lymphocytes in the pineal gland has been shown to regulate the production of melatonin in a circadian manner, establishing a direct feedback loop between the immune system and the pineal gland. This highlights the bidirectional communication and regulation between the immune system and a gland that is known for its immune regulatory functions.

## 6. Conclusions

The results of this study illustrate that different colours and light intensities have effects on the development and immune function of lymphoid organs, all of which are mediated by melatonin activity. The data reported in this paper are in agreement with our results, which will be published in the near future.

## Figures and Tables

**Figure 1 animals-13-02095-f001:**
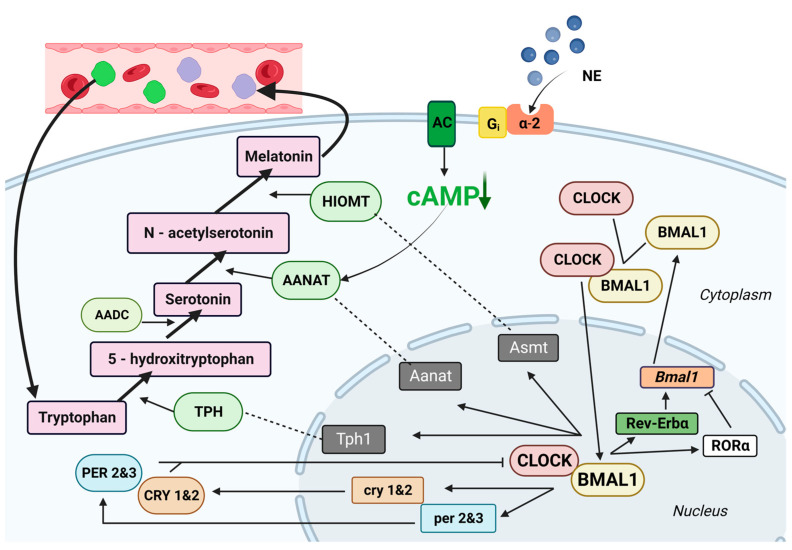
Schematic representation of melatonin biosynthesis in the epiphysis. The process of melatonin synthesis starts from the amino acid tryptophan, which is taken up from the blood circulation and converted into 5-hydroxytryptophan and then into serotonin. During the night, AANAT (arylalkylamine N-acetyltransferase) activity increases and helps convert serotonin to N-acetylserotonin, which is further converted to melatonin by hydroxyindole-O-methyltransferase (HIOMT). Eventually, melatonin is distributed to other organs via the blood circulation. NE (norepinephrine), AC (adenylate cyclase), α-2 (receptor), cAMP (cyclic adenosine monophosphate), arylalkylamine N-acetyltransferase (AANAT), aromatic L-amino acid decarboxylase (AADC), hydroxide-O-methyltransferase (HIOMT), circadian locomotor output cycles kaput (CLOCK), brain and muscle aryl hydrocarbon receptor nuclear translocator-like protein 1 (BMAL1), PER 1 and 3 (period), CRY 1 and 2 (cryptochrome). The figure was created with www.BioRender.com (accessed on 7 June 2023).

**Figure 2 animals-13-02095-f002:**
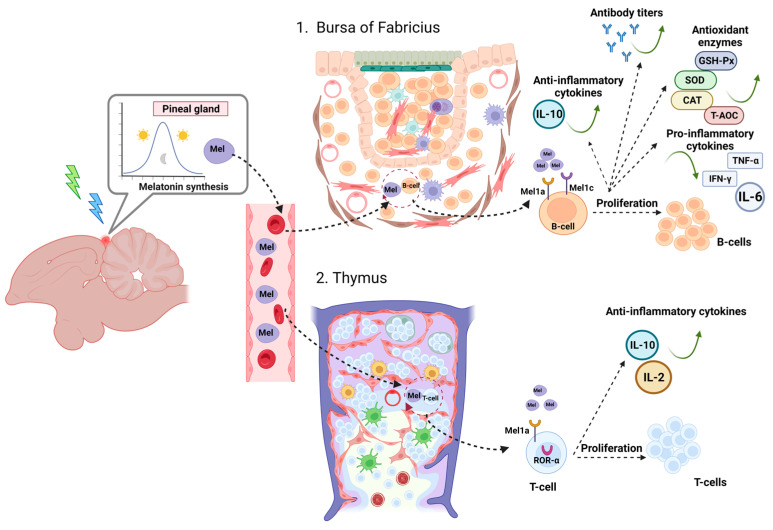
Schematic representation of how melatonin modulates the activity of the bursa of Fabricius and thymus induced by the combination of green and blue monochromatic light in chickens. A combination of green and blue monochromatic light increased serum melatonin concentration. In the bursa of Fabricius, the interaction between B lymphocyte and Mel (melatonin) via the membrane receptors Mel1a and Mel1c resulted in the proliferation of B lymphocytes and an increased antibody titres; a reduced oxidative stress via the increased concentrations of the antioxidant enzymes CAT, SOD, T-AOC, and GSH-Px; an increased secretion of anti-inflammatory cytokine IL-10; and the concurrent decrease in the secretion of pro-inflammatory cytokines IFN-γ, TNF-α, and IL-6. In the thymus, via membrane (Mel1a) and nuclear (ROR-α) receptors, melatonin (Mel) led to T lymphocyte proliferation and secretion of the anti-inflammatory cytokines IL-10 and IL-2. The figure was created with www.BioRender.com (accessed on 8 June 2023).

**Figure 3 animals-13-02095-f003:**
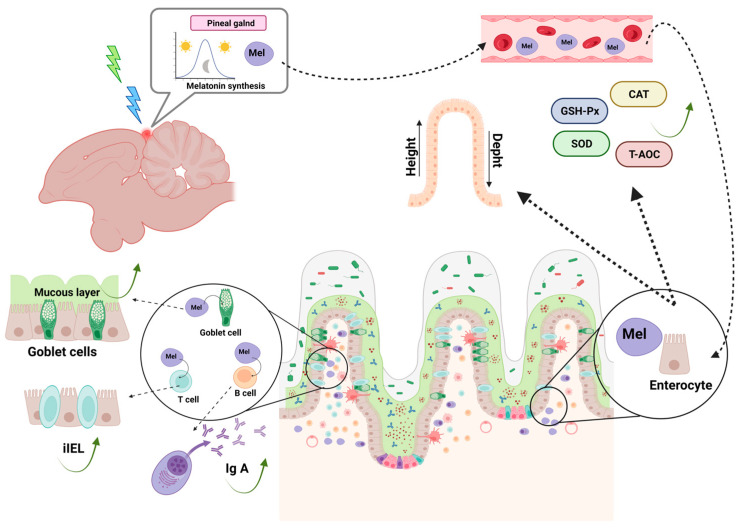
Schematic representation of how melatonin modulates secondary lymphoid organ (GALT) activity induced by the combination of green and blue monochromatic light in chickens. With the help of blood circulation, melatonin (Mel) reaches the digestive tract, where it will induce strong local immunity via: (1) an increase in the height of intestinal villi, increase in the depth of intestinal crypts, and increase in the villi/crypt ratio; (2) the maintenance of an antioxidant state by increasing CAT, SOD, T-AOC, and GSH-Px levels; (3) an increase in the protective mucus layer; and (4) a proliferation of intraepithelial lymphocytes (iIEL) and Ig A levels. The figure was created with www.BioRender.com (accessed on 8 June 2023).

## Data Availability

Not applicable.

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
