# Peer review of "Influence of Different Light Spectra on Melatonin Synthesis by the Pineal Gland and Influence on the Immune System in Chickens"

_animals, 2023, doi:10.3390/ani13132095_

Round 1
Reviewer 1 Report
Very well organized and comprehensive review. The review addresses the focus on the effect of light on immune system and the welfare of poultry and gives interesting and well explained information on the correlation.
The topic is not original but the point of view is innovative considering that large manufactories operating in the sector of "light" are developing new lighting systems for the poultry house. The authors describe logically and clearly all the possible influences of light on melatonin production and immune system effects.
The work is a review, in my opinion the different studies and results are well presented. The cited studies and the argument itself are very relevant and as many stakeholders and light industries are experimenting innovations in this field. The conclusions are consistent with the evidence and arguments presented. The references are appropriate.
Figures could be improved, in addition the work would be more complete representing better every reception mechanism anatomically and physiolocically speaking. Maybe more figures related to chicken anatomy and physiology and light receptors.
Just two minor comments:
Line 37 add after pineal gland: (or epiphysis)
Line 348 erase (prolong the antibody effect time) as is already said
Author Response
Dear Reviewer,
Please see the attachment with all changes.
Best regards,
Loredana Horodincu
Doctor- Department of Preclinics, Faculty of Veterinary Medicine, Iasi
University of Life Sciences; loredana.horodincu@uaiasi.ro

Reviewer 2 Report
The manuscript submitted by Loredana Horodincu and Carmen Solcan titled: “Influence of different light spectra on melatonin synthesis by the pineal gland and influence on the immune system in chickens” concerns an interesting subject, and in my opinion after major revision can be published in the Animals journal.
GENERAL COMMENTS
The term epiphyseal gland should be changed to pineal due to the very infrequent use of that term.
The article suggests the existence of a new regulatory axis, but this axis has long functioned in the literature under a different name, details below.
The biological and molecular clock located e.g., in the pineal gland of birds has been practically omitted in the text.
The work completely omitted the issue of the influence of the stimulated immune system on the biosynthetic activity of the pineal gland.
One of the main elements of the biological clock, i.e., the suprachiasmatic nuclei (SCN) of the hypothalamus, is not precisely referred in the text.
The text does not pay attention to the Pineal Associated Leukocyte Tissue (PALT) inside of the avian pineal gland.
It should be mentioned that NE is not necessary for the regulation of melatonin biosynthesis in birds, and daily changes in the activity of enzymes involved in melatonin biosynthesis are controlled by molecular clock ticking in pinealocytes. All gene encoding enzymes involved in melatonin biosynthesis are clock controlled genes (CCGs).
It should be mentioned in the text that leukocytes release melatonin, which acts on auto- and paracrine principles.
In my opinion on the basis of the publication, "Role of monochromatic light on development of cecal tonsil in young broilers: light colour and cecal tonsil development of chick" far-reaching conclusions should not be reached (section 4). To my knowledge, the results presented in this publication have not yet been confirmed by other research groups. In addition, changes in the immunological parameters measured in this study are very small and rarely statistically significant.
SOME DETAILED COMMENTS AND SUGGESTION
Line10 - I suggest using the term pineal gland instead of the epiphyseal gland, because the term epiphyseal gland used in the first sentence of the paper is a misnomer and is very rarely used.
Line 11 – “The purpose of this research is to point out the endocrine-immune correlations between melatonin and lymphoid organ activity in broilers” – this sentence is misleading; the paper presents data derived from many species of birds and mammals too.
Line12 - “between melatonin and lymphoid organ activity in broilers” I propose - between melatonin and immune system activity in broilers.
Line 12 - 14 ; “the axis formed by: light, epiphyseal gland, lymphoid organs and immune response” – it is not necessary to enter a new name for a long-existing one. In the existing literature the corresponding term is “Immune-Pineal Axis”.
Line 141 – “visinin” – lack of introduction on what kind of protein it is.
Line 171 – “the elimination of free radicals from the body” – I suggest using term e,g. free radical scavenger.
Line 198 - Figure 1 is an overly simplified scheme of melatonin biosynthesis, because AANAT is not the only enzyme that affects melatonin biosynthesis. Moreover, the expression of other enzymes involved in melatonin biosynthesis is also regulated by the molecular clock ticking in the pinealocytes. This information does not exist in the main text as well as on the figure.
Line 396 – thymus instead of “timus”.
Line 429 – 431 – My proposal is: RORα could negatively regulate green light (560 nm)-induced T lymphocyte proliferation in the chick thymus by regulating IκB phosphorylation, which promotes RELA (synonym p65) nuclear translocation as a main effector of NF-κB intracellular pathway [99].
Line 436 – “increasing the expression of p-IκB” I suggest: increasing the level of phosphorylated IκB.
Line 431 – the name of the P65 protein should be changed to RELA - this is the correct nomenclature name (HGNC: www.genenames.org). I also suggest describing elements involved in NF-κB pathway activation more clearly such as: IκB, IκBα.
Line 459 - IL2 and symbols of other cytokines should be corrected. Abbreviation IL2 defines the gene name, appropriate symbol for protein is IL-2.
Line 463 - The Figure 2 is not legible, but the names of cytokines are presented in the correct nomenclature.
QUESTIONS WICH SHOULD BE ANSWERSD IN THE TEXT
Does the melatonin produced by the digestive system have no effect on the presented relationships?
Does experimental data indicating that melatonin released by the pineal gland increases melatonin levels in the digestive system exist?
The text submitted for review should undergo professional language editing.
Author Response

(The authors gave the same response as above.)
